# Pathways from Neuroticism, Social Support, and Sleep Quality to Antenatal Depression during the Third Trimester of Pregnancy

**DOI:** 10.3390/ijerph19095602

**Published:** 2022-05-05

**Authors:** Jiarui Chen, Mei Sun, Chongmei Huang, Jinnan Xiao, Siyuan Tang, Qirong Chen

**Affiliations:** 1Xiangya School of Nursing, Centreal South University, 172 Tongzipo Road, Changsha 410013, China; chenjr@cus.edu.cn (J.C.); smnjw2008@126.com (M.S.); huangcm@csu.edu.cn (C.H.); jnxiao2021@csu.edu.cn (J.X.); tsycongcong@126.com (S.T.); 2Hunan Women’s and Children’s Health and Development Research Center, Changsha 410013, China

**Keywords:** neuroticism, social support, sleep quality, antenatal depression

## Abstract

*Background*: Antenatal depression is a severe public health problem. Many studies support the concept that neuroticism, social support, and sleep quality are closely related to antenatal depression. However, there is little evidence concerning the influencing pathways of these variables on antenatal depression. The aim of this study is to investigate the pathways from neuroticism, social support, and sleep quality to antenatal depression during the third trimester of pregnancy. *Methods*: A cross-sectional study design was used. A total of 773 eligible women in the third trimester of pregnancy submitted valid questionnaires from June 2016 to April 2017. Instruments with good reliability and validity were used to measure neuroticism, social support, sleep quality, and antenatal depression. Structural equation modeling was used to explore the pathways from neuroticism, social support, and sleep quality to antenatal depression during the third trimester of pregnancy. *Results*: Antenatal depression is shown to be positively correlated with neuroticism and negatively correlated with social support and sleep quality. Neuroticism is shown to have a direct effect and indirect effects through social support and sleep quality on antenatal depression. *Conclusions*: Neuroticism influences antenatal depression directly and indirectly. Social support and sleep quality are the mediators of the indirect relationship between neuroticism and antenatal depression. Our results suggest that a personality test offered to all pregnant women could help detect a vulnerability to depression, whereupon intervention in the domains of sleep and social support could prove preventive.

## 1. Introduction

Depression is a common mental health disorder and is the main cause of disability worldwide. The World Health Organization reported that about 10% of pregnant women and 13% of perinatal women worldwide have mental disorders, mainly depression [1]. A meta-analysis that included 296,284 women with maternal depression in 56 countries showed that the global prevalence of depression during pregnancy and childbirth was about 17.7%. Research conducted in China showed that the prevalence of depression during pregnancy ranges from 15% to 30% [2,3]. Some studies showed that the prevalence of antenatal depression was higher in the third trimester of pregnancy compared to other trimesters of pregnancy [4,5]. 

Depression during pregnancy has attracted the attention of researchers not only because of its high prevalence but also on account of the various effects it has. Antenatal depression can affect the woman herself, the fetus, as well as the whole family, and even society [6,7,8]. For pregnant women themselves, the physical and psychological symptoms of depression can affect many aspects of their daily life; these symptoms can include an unhealthy diet and malnutrition [9,10], sleep disorders [11], poor quality of life, reduced possibility of regular health care during pregnancy [12], and even suicide [13]. Depression during pregnancy also correlates with the mental health of their spouses. Studies have found that husbands with a depressed wife are at an increased risk of anxiety, depression, and decreased marital satisfaction [14] during the pregnancy and postpartum of their wives [15]. In addition, depression during pregnancy has a great impact on the fetus. It was reported that fetal development delay and reduced fetal activities may be associated with depression during pregnancy [7,8]. In addition, antenatal depression can lead to risky behaviors, childcare difficulties, and social burdens [16].

Many studies have explored the risk factors of antenatal depression to provide evidence that can be used in the development of antenatal depression interventions. In many previous studies, neuroticism has shown a significant correlation with antenatal depression [17,18]. Neuroticism has been variously defined as emotional instability, negative outlook, difficulties in adjustment, poor self-control, a tendency to complain about life, and an inability to cope with psychological stress [19]. This construct, which is no longer in worldwide use, is traditionally measured by the Eysenck Personality test [20]. Pregnant women with a tendency for neuroticism have a higher risk of antenatal depression. A study concerning neuroticism and perinatal depression among late-pregnancy women showed that while controlling for confounding factors, non-depressed women with high neuroticism scores were four times more likely to develop symptoms of depression than those with low neuroticism scores [21]. As it is a personality trait, neuroticism is difficult to change. Where intervention is helpful is in the connection between neuroticism and depression, especially in pregnancy.

Interventions can be directed toward increasing social support and improving sleep quality as both relate to neuroticism and depression [22]. It has been reported that neuroticism affects the effectiveness of social support acceptance, resulting in a reduction in perceived social support [23]. Neuroticism also negatively affects sleep quality [24]. Meanwhile, social support and sleep quality are important influencing factors in antenatal depression [25,26]. Furthermore, social support is also related to sleep quality [27]. We sought to determine whether social support and sleep quality mediate the relationship between neuroticism and antenatal depression, as this could provide evidence for the development of effective interventions in antenatal depression. No previous studies have addressed this question. 

### Aim and Hypotheses

As the third trimester of pregnancy is a period in which women are at a high risk of antenatal depression, this study aimed to explore the pathways from neuroticism, social support, and sleep quality to antenatal depression in the third trimester of pregnancy. Based on the evidence from the previous literature, we hypothesized that the following: (1) neuroticism is directly and indirectly related to antenatal depression; (2) social support and sleep quality are mediators of the relationship between neuroticism and antenatal depression. Figure 1 shows the study’s predefined path model in this study. This study provides more evidence that can be used in the prevention and treatment of antenatal depression during late pregnancy.

## 2. Methods

### 2.1. Study Design

This study used a cross-sectional design. This study was reported following the Strengthening the Reporting of Observational Studies in Epidemiology checklist (STROBE) to improve the reporting quality of this study (more details can be found in the Appendix A).

### 2.2. Setting and Participants

First, the researchers randomly selected three general tertiary hospitals as research settings from all 21 general tertiary hospitals in Changsha, China, using a simple random sampling method. In these three hospitals, the investigators recruited 900 pregnant women who came to the hospitals for regular prenatal care at the obstetrics clinics. The inclusion criteria for participants were the following: (1) pregnant woman; (2) with an intrauterine pregnancy; (3) in the third trimester of pregnancy; (4) able to understand the purpose and process of the research and complete the questionnaire independently. The exclusion criteria for participants were the following: (1) having a severe mental illness diagnosed by a psychiatrist; (2) having an intellectual disability, cognitive impairment, or brain damage diagnosed by a psychiatrist or clinician; (3) participating in any psychological intervention. There are many rules for sample size calculation in the structural equation model. As a commonly used rule, a sample size of at least 200 is recommended for structural equation model analysis, and it is also suggested that the sample size should be more than the value of (5–10 cases) × (the number of free parameters) [28]. There were 16 free parameters to be estimated in this model, so a sample size of more than 90–160 was required. The sample size in this study met the criteria above.

### 2.3. Recruitment and Data Collection

A total of 900 questionnaires were distributed to eligible pregnant women from June 2016 to April 2017. In total, 827 participants returned questionnaires. Questionnaires with less than 75% completed items were removed, leaving a total of 773 valid questionnaires. The investigators provided all of the information regarding the research to all of the potential participants in the obstetrics clinics, including an explanation of the objectives and the investigation process of the research, anonymity, and voluntary participation. After oral informed consent was obtained, the questionnaires were delivered to those who were willing to participate. Upon completion, the participants returned the questionnaires to the investigators. The investigator checked the completed questionnaires and asked participants to add information if there were missing data. The data were collected from June 2016 to April 2017. 

### 2.4. Instruments

#### 2.4.1. Eysenck Personality Questionnaire-Neuroticism (EPQ-N)

The Eysenck Personality Questionnaire is a self-reporting scale that was developed by the British psychologist H. J. Eysenck [29]. The English version of the EPQ adult questionnaire has a total of 107 items, and the Chinese version of the EPQ was revised by Chen [30]. It includes four subscales with a total of 85 items. Based on the finding from the literature review that the personality trait of neuroticism is related to the occurrence of perinatal depression, this study only used the neuroticism subscale (EPQ-N) in the EPQ questionnaire to measure the neuroticism characteristic. There are 24 entries in the EPQ-N. Answers form a dichotomy, with one point corresponding to “yes” and zero points corresponding to “no”. Cronbach’s α of the Chinese version of the EPQ-N was found to be 0.771 in a study conducted in China [30]. The scores of all of the items are added up to obtain a total score, and then, the total score is converted into a standard T score based on age and gender norms. For neuroticism, high scores are characterized by constant worry, tension, anxiety, depression, mood swings, being prone to strong emotional reactions, and even irrational behaviors when encountering stimuli. Individuals with low EPQ-N scores are characterized by a tendency to have slow and mild emotional reactions and can easily become calm again even if their emotions are aroused. Usually, people with lower scores are more stable, gentle, and have better self-control capacity.

#### 2.4.2. Social Support Revalued Scale (SSRS)

The social support assessment scale used in this study was developed by Dr. Xiao [31]. It has been widely used in the measurement of social support for different groups in China since 1990. The scale contains ten items involving three dimensions objective support (3 items), subjective support (4 items), and support utilization (3 items). Objective support means the objective and visible support reported by the participant, including financial support and the existence of participation in social networks and group relations. Subjective support refers to an individual’s emotional experience of being respected, supported, and understood by their families, friends, and society. Considering this is a self-reported scale, objective support and subjective support could represent perceived support. Support utilization refers to the interaction between the individual and the social support system. The use of existing support and the provision of support to others are all forms of support utilization. Previous studies have shown that SSRS has good reliability and validity. Cronbach’s α for the three dimensions and the composite scale ranges from 0.825 to 0.896 [32]. Additionally, the correlation coefficients between the three subscales and the total scale range from 0.724 to 0.835, indicating that the scale has good content validity [33].

#### 2.4.3. Pittsburgh Sleep Quality Index (PSQI)

The PSQI was compiled by Dr. Buysse of the University of Pittsburgh in 1989 [34]. The scale was originally used to evaluate the sleep quality of patients with sleep disorders and mental disorders, and later studies found that it is also applicable to the evaluation of the sleep quality of individuals without sleep or mental disorders. The scale consists of 4 blank questions and 5 choice questions, in which one choice question contains 10 shorter questions. The PSQI is used to assess the quality of an individual’s sleep in the last month. The total score ranges from 0 to 2 l. The higher the score, the worse the sleep quality. This scale has been proven to have good reliability and validity in the evaluation of the sleep quality of pregnant women [35]. Researchers translated it into Chinese, and Cronbach’s α coefficient of the scale was 0.734 in Chinese pregnant women [36,37]. The Chinese version of PSQI has a sensitivity of 88% and a specificity of 84% [38].

#### 2.4.4. Edinburgh Postnatal Depression Scale (EPDS)

The EPDS is a 4-point Likert scale containing a total of 10 items. Each item scores 0–3 based on the severity of the symptoms. It was developed by Cox in 1987 [39] and was originally used as an effective screening tool for postpartum depression by researchers and perinatal health caregivers. Because of its high sensitivity and specificity for perinatal depression screening and the fact that it has fewer items and is easy to score, it has also been widely used in perinatal depression screening worldwide. The Chinese version of the EPDS used in this study was translated by Lee et al. in 1998 [40]. In the EPDS, the recommended threshold for possible depression is 9, and those with scores less than 9 are considered to not have depression; those with a score of 9 or greater are considered likely to have depression. The Chinese version of the EPDS has a sensitivity of 82%, a specificity of 86%, a positive predictive value of 44%, and a negative predictive value of 97%. Its Cronbach’s α coefficient is 0.762 [41,42].

### 2.5. Statistical Methods

Statistical Product and Service Solutions (SPSS) version 25.0 was used for descriptive analysis and correlation analysis between variables. Analysis of moment structure (AMOS) version 25.0 was used for structural equation modeling analysis. In total, 54 questionnaires with more than 25% missing data were regarded as invalid questionnaires and removed. There were 42 questionnaires with missing data (the number of missing values ranged from 5 to 10) in the 773 valid questionnaires. The missing data from the included questionnaires were supplemented according to the corresponding average value. In the structural equation modeling analysis, to ensure the validity of the results, we also made the data analysis using the data set without supplementary data. In the structural equation model, social support was divided into two variables, i.e., perceived support and support utilization. Perceived support was regarded as a latent variable and was formed by two manifest variables (i.e., objective support and subjective support). In this case, the relationship between manifest variables (i.e., objective support and subjective support) and the latent variable (perceived support) is formative. Other manifest variables (neuroticism, support utilization, sleep, and antenatal depression) were indicated using the EPQN score, support utilization score, PSQI score, and EPDS score, respectively, which were all measured with a questionnaire that only included one dimension. The maximum likelihood method was chosen to estimate the covariance matrix, and Bootstrap to 2000 re-entries was used to test the direct and indirect effects of the variables. The following indices were used to evaluate the goodness-of-fit of the structural equation model: chi-square (*χ*^2^), degrees of freedom (*df*), the chi-square/degrees of freedom ratio (*χ*^2^/*df*), the incremental fit index (IFI), the comparative fit index (CFI), the Gamma goodness-of-fit index (GFI), the adjusted goodness-of-fit index (AGFI), the normal fit index (NFI), and the root mean square error of approximation (RMSEA). A well-fitted model requires the IFI, CFI, GFI, AGFI, NFI ≥ 0.90, RMSEA ≤ 0.05, and 1 < *χ*^2^/*df* < 3.

### 2.6. Ethical Approval

This study was approved by an Institutional Review Board (IRB) on 20 May 2016. All participants were provided with full information about this study.

## 3. Result

A total of 900 questionnaires were distributed during the recruitment. In all, 773 valid questionnaires were returned. The response rate was 85.9%. In total, 229 women had an EPDS score of greater than 9, indicating that 29.6% of people in this group may have been at risk of antenatal depression. The scores of each scale are shown in Table 1. General information regarding the participants can be found in Table 1 of a previously published paper [43]. (It is noted that the main data used in these two studies are different.)

### 3.1. The Correlation between Variables

Correlation analysis was performed using the scores of perceived support, support utilization, sleep quality, neuroticism, and antenatal depression. The correlation coefficient values (*r*) of 0.10–0.29, 0.30–0.49, and 0.50–1 suggest a small, moderate, and strong correlation, respectively. The results showed that neuroticism and perceived support were highly related to antenatal depression with a positive correlation (*r* = 0.603, *p* < 0.01) and a negative correlation (*r* = −0.514, *p* < 0.01), respectively. Furthermore, neuroticism was positively correlated to sleep quality (*r* = 0.331, *p* < 0.01) and negatively correlated to perceived support and support utilization (*r* = −0.436, *p* < 0.01 and *r* = −0.193, *p* < 0.01, respectively). It is noted that higher PSQI scores indicate worse sleep quality. Sleep quality was negatively related to perceived support and support utilization, while perceived support was positively related to support utilization. More details can be seen in Table 2.

### 3.2. The Pathways from Neuroticism, Social Support, and Sleep Quality to Antenatal Depression

The fit coefficients of the final model were as follows: *χ*^2^ = 12.813, *d*ƒ = 5, 1 < *χ*^2^/*d*ƒ = 2.563 < 3, IFI = 0.992 > 0.9, CFI = 0.992 > 0.9, GFI = 0.995 > 0.9, AGFI = 0.977 > 0.9, NFI = 0.987 > 0.9, and RMSEA = 0.045 < 0.05. These results suggest that this model could be accepted as a well-fitted model. Neuroticism was shown to have direct and indirect effects on antenatal depression, while sleep quality and perceived support were shown to have a direct effect, and support utilization only had an indirect effect on antenatal depression. Neuroticism had a large effect (coefficient = 0.556, *p* < 0.01) on antenatal depression. Furthermore, the neuroticism also indirectly influenced (coefficient = 0.263, *p* < 0.01) antenatal depression through social support (i.e., perceived support and support utilization) and sleep quality. This means that perceived support and sleep quality were the mediating variables in the relationship between neuroticism and antenatal depression. Furthermore, support utilization played a role in the influence of neuroticism on perceived support and sleep quality, which were the two mediators in the relationship between neuroticism and antenatal depression. More details can be found in Table 3 and Figure 2. Furthermore, the data analysis results using the data set without supplementary data also support the results of the data analysis using the data set with supplementary data.

## 4. Discussion

Our study shows that the EPDS average score among women in the third trimester of pregnancy was 6.76 ± 3.042, and 29.6% of the participants scored 9 points or more on the EPDS. The results were similar to other studies concerning the same ethnic groups [44]. The standard scores of the EPQ-N among the participants in this study were between 32.90 and 78.40, with a mean of 50.00 (*SD* = 10.000), which indicated that, in general, the sample in this study did not tend to have a high score in the EPQ-N [45]. The relationships found between neuroticism, social support, sleep quality, and antenatal depression allow us to confirm our hypothesis that neuroticism not only directly influences depression during pregnancy but also affects it through perceived support and sleep quality. Furthermore, support utilization plays a role in the influence of neuroticism on perceived support and sleep quality, which are the two mediators in the relationship between neuroticism and antenatal depression. Therefore, social support (perceived support and support utilization) and sleep quality are the mediators between neuroticism and antenatal depression. 

Interest in the relationship between personality traits and maternal depression has increased in recent years. A study concerning neuroticism and depression showed that the presence of higher scores in neuroticism could increase the probability of a major depressive episode after a major biological, psychological, and social life event such as giving birth [18]. A recent systematic review of personality traits and postpartum depression found 34 studies, of which 17 studies indicated that higher scores in neuroticism were a risk factor for postpartum depression [46]. People with high neuroticism scores may show unstable emotions. They often perceive themselves as being ineffective at coping and engaging in worrying, rumination, or emotional avoidance [47]. Therefore, it would be more likely for women with higher levels of neuroticism to experience intense negative emotions in response to pregnancy and childbirth. Based on the results of our study, neuroticism can directly influence antenatal depression and indirectly affect antenatal depression through perceived support, support utilization, and sleep quality. In the structural equation model of our study, the latent variable—perceived support—was formed by two measured variables—objective support and subjective support, which were measured using two subscales of the self-report questionnaire—the *Social Support Revalued Scale (SSRS)*. Some studies have also pointed out that neuroticism affects both the provision and acceptance of social support, especially social support acceptance, thereby reducing perceived support [48,49]. Considering the definition of neuroticism, we may propose that due to their unstable personalities, sensitivity, and the ease with which they experience negative emotions, pregnant women with higher scores in EPQ-N may feel less social support and be less likely to utilize support and provide support to others, which can affect their mental health, causing antenatal depression. Furthermore, sleep quality has been shown to have a negative impact on depression in previous studies [50]. 

Our study shows that social support was inversely associated with antenatal depression, which is consistent with the results of existing studies [51]. In our study, social support included perceived support and support utilization. Perceived support was shown by self-reported objective support and subjective support. Objective support refers to objective and visible support reported by the participants, including financial support and the existence of and participation in social networks and group relations. Additionally, subjective support concerns an individual’s emotional experience of being respected, supported, and understood by their family, friends, and society. The results from our study confirmed that perceived support was highly correlated with antenatal depression, which is consistent with existing evidence [52]. Pregnancy and childbirth are stressful events for women, especially for those with higher scores on the EPQ-N. They tend to have more negative emotions during pregnancy. During this period, social support received from a spouse, family, and/or friends can help pregnant women emotionally and spiritually and relieve their negative emotions, which may prevent them from experiencing antenatal depression. 

In our study, social support and sleep quality were fully identified as two mediating variables in the pathway from neuroticism to antenatal depression. This study also pointed out that support utilization also affected the pathway from neuroticism to perceived support and sleep quality. Support utilization refers to the interaction between an individual and a social support system. It includes both the use of existing support and the provision of support to others [31]. For women with higher scores in EPQ-N, better social support utilization can increase their perceived support, thereby improving their ability to cope with stressors in pregnancy and reducing the risk of antenatal depression. Meanwhile, better support utilization can also help to reduce the incidence of negative emotions, thereby improving sleep quality [53]. Therefore, for pregnant women who have high scores in EPQ-N, it is important to improve their social support networks. The following several methods can be adopted: (1) improving the objective and subjective support provided to pregnant women; (2) helping pregnant women identify support; (3) helping pregnant women utilize support.

### Limitations

This study was a cross-sectional design study conducted in Changsha, China. To acquire enough eligible participants, we recruited participants in tertiary hospitals (these tertiary hospitals were selected using a simple random sampling method). However, this may affect the representativeness of the sample. Furthermore, this study focused on pregnant women in the third trimester of pregnancy. Therefore, caution is needed when expanding the conclusions in this study to other populations (e.g., pregnant women in other cultures or countries, pregnant women attending medical institutions at lower levels rather than tertiary hospitals, pregnant women in the first trimester of pregnancy, etc.). Future studies that include participants in other trimesters of pregnancy, from other cities, cultures, and countries, and recruited from different types of medical institutions are required. The data used in this study were self-reported, which is also a limitation of this study, even though we used instruments with good reliability and validity.

## 5. Conclusions

To our knowledge, this is the first study to explore the pathways between neuroticism, social support, and sleep quality to antenatal depression during the third trimester of pregnancy. Neuroticism is a significant risk factor for antenatal depression. As a personality characteristic, neuroticism is difficult to change. Therefore, the exploration of the changeable variables that mediate the relationship between neuroticism and antenatal depression is crucial in order to improving the mental health of pregnant women with a higher level of neuroticism. This study suggests that social support (i.e., perceived support and support utilization) and sleep quality are the important mediators between neuroticism and antenatal depression. In conclusion, we recommend offering brief personality tests during antenatal assessments and, should neuroticism scores be high, addressing sleep and social support in order to prevent depression.

## Figures and Tables

**Figure 1 ijerph-19-05602-f001:**
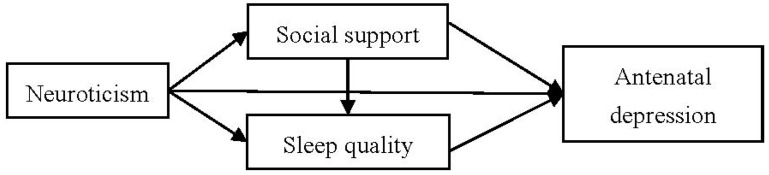
The predefined path model (pathways from neuroticism, social support, and sleep quality to antenatal depression during the third trimester of pregnancy).

**Figure 2 ijerph-19-05602-f002:**
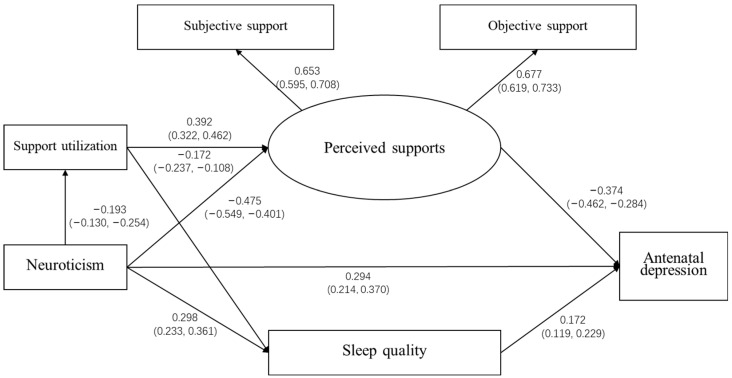
The standardized path model (pathways from neuroticism, social support, and sleep quality to antenatal depression during the third trimester of pregnancy). *Note*: (1) Sleep quality was assessed using the Pittsburgh sleep quality index (PSQI). A higher score of the PSQI means worse sleep quality.

**Table 1 ijerph-19-05602-t001:** Descriptive statistics of neuroticism, social support, sleep quality, and antenatal depression (*n* = 773).

	Minimum	Maximum	$\overset{¯}{X}\;\pm \;SD$
Neuroticism ^a^ (EPQ-N score)	32.90	78.40	50.00 ± 10.000
Social support (SSRS score)	24.00	58.00	41.68 ± 6.708
Perceived support	19.00	48.00	33.73 ± 5.955
Support utilization	4.00	12.00	7.95 ± 1.537
Sleep quality (PSQI score)	1.00	18.00	7.15 ± 3.102
Antenatal depression (EPDS score)	0	15.00	6.76 ± 3.042

Note: ^a^: The score of neuroticism is the standard score of EPQ-N.

**Table 2 ijerph-19-05602-t002:** Correlations between neuroticism, social support, sleep quality, and antenatal depression (*n* = 773).

	Antenatal Depression	Neuroticism	Sleep Quality	Social Support	
				Perceived Supports	Support Utilization
Antenatal depression	1	0.603 **	0.412 **	−0.514 **	−0.291 **
Neuroticism		1	0.331 **	−0.436 **	−0.193 **
Sleep quality			1	−0.261 **	−0.229 **
Perceived support				1	0.337 **
Support utilization					1

Note: ** *p* < 0.01.

**Table 3 ijerph-19-05602-t003:** Effects of neuroticism, social support, and sleep quality on antenatal depression (*n* = 773).

		Neuroticism	Sleep Quality	Social Support	
				Perceived Supports	Support Utilization
Antenatal depression	Direct effect	0.294 **	0.172 **	−0.374 **	0.000
	Indirect effect	0.263 **	0.000	0.000	−0.176 **
	Total effect	0.556 **	0.172 **	− 0.374 **	−0.176 **

Note: ** *p*<0.01.

## Data Availability

The data that support the findings of this study are available from the corresponding author upon reasonable request.

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
