# Peer review of "Pathways from Neuroticism, Social Support, and Sleep Quality to Antenatal Depression during the Third Trimester of Pregnancy"

_ijerph, 2022, doi:10.3390/ijerph19095602_

Round 1

Reviewer 1 Report

The manuscript "Pathways from neuroticism, social support, and sleep quality to antenatal depression during the third trimester of pregnancy" has the quality of its english significantly improved when compared with the previous version.

Comments

1) Lines 110-112 it is not clear how the information for exclusion was actually checked for each enrolled subject. Who diagnosed the mental status? How homogeneous was that evaluation?

2) The study has an extremely poor design not even acknowledged as a major  limitation. Major confounders such as age, and extended socio economic indicators were not considered.

3)Lines 193-194 not clear what software was used for which analysis.

4) Lines 195-197  seems there was quite a lot of data inputation considering samples with up to 25%  of missing answers were considered. How different are the results when considering only samples with all relevant responses? This test is essential to support the validity of results. 

5) Lines 217-223 are extremely mystified. What does and how does each one of the tests check model goodness of fit? AGain how different would be the results without missing/inputed data?

6) Table 2 info would be better presented as a correlation plot.

7) Lines 251-252 : Related to 5 how do you interpret the different goodnes of fit metrics?

Reviewer 2 Report

Thank you for revising the paper with my suggestions.

I think that now your work is much improved.

I suggest you only one little but important insight: In Table 3 and in the results section you must insert the p.value. In Table 3 you should consider to add two columns for Lower and Upper Confidence Intervals. See Signore, F., Pasca, P., Valente, W., Ciavolino, E., & Ingusci, E. (2021). SOCIAL RESOURCES AND EMOTIONAL EXHAUSTION: THE ROLE OF COMMUNICATION IN PROFESSIONAL RELATIONSHIPS. Intellectual Economics15(2).

In Figure 2 the non-significant structural relationship can be represented with dotted arrows and the significant ones with continue arrows. This could facilitate the interpretation.

Round 2

Reviewer 1 Report

Changes were not satisfactory and most comments were ignored.

Author Response

This manuscript is a resubmission of an earlier submission. The following is a list of the peer review reports and author responses from that submission.

Round 1

Reviewer 1 Report

The manuscript "Pathways of neuroticism, social support, and sleep quality to antenatal depression during the third trimester of pregnancy" is written in an extremely poor quality english, something that makes difficult its evaluation. In the next lines I provide feedback on what i was able to understand but any serious consideration requires the manuscript to be professionally edited. For example, just starting by the title, where it says "... of neuroticism ..." should say "... from neuroticism ..." and the manuscript is plagues with such unacceptable language problems.

Comments

 1)The manuscript claims to have followed STROBE recommendations but no details about the date of approval and how does relate to the study dates is presented in the ms.

2) The introduction is not understandable and written in an unintelligible fashion. The goals/aims are not clearly presented and the path analysis is not well justified nor connected with the background info in.

3)Methods the language is again unintelligible, and the dates and basic data about the studied population are not reported. Authors should seriously check STROBE and articles in the area to see the expectations for reporting recruitment of subjects and report how many individuals were not recruited how did they check the statistical power for their final sample size, etc.

4) The results section is also written in an unintellegible fashion. Table 1 has issues as authors do not report how many individuals were in each class and basic details that should be reported in this kind of analyses. The path model is not explained and just fitiing results are reported but not minimally interpreted.

5) The discussion and conclusion sections are also poorly written and little effort was made to compare results with other studies. On the positive side authors try to assess the limitations of the study.

Reviewer 2 Report

Dear authors,
thank you for the opportunity to review your article on the role of certain personality traits, sleep quality and social support in pre-partum depression. The paper is not very methodologically clear and correct, however I suggest some major revisions to be considered for its improvement:
1) In the abstract, anticipate which measures you used to define measures "with good psychometric properties".
2) I would use more formal language, not "We should pay more attention".
3) Include in the abstract the period of completion of the questionnaire.
4) After the introduction you should insert a sub-section called "Aims and Hypotheses" in which you explain the objectives of the paper.
5) Figure 1 must be centred and above all adjusted. Certain things should be checked before submitting articles.
6) Line 99. What is the meaning of the phrase "A cross-sectional design study."?
7) You should include for ALL questionnaires used measures of reliability and AT LEAST one measure of validity, such as the AVE, as well as AT LEAST one example item. Moreover, how many indicators you used for each latent variable? And what kind of relationship between manifest and latent variables you adopted (reflective or formative)?
8) Line 182: bootstrap to how many re-entries?
9) Throughout the article choose whether to use two decimal places or three and make it uniform, not sometimes two decimal places and sometimes three.
10) My God, figure 2 is illegible. Please modify it because I don't understand anything!

11) Nothing about the mediation? Actually in your model there are two mediation. Please explain the type of mediation obtained.
12) The discussions should recall the hypotheses, writing "the relation between ... and ... allows to confirm / not confirm hypothesis 1" etc etc.
13) Limitations: the fact that the measures are self-report is missing.
14) In Table 1 insert the asymmetry and kurtosis indices to justify the use of SEMs.

15) In one of the two figures, you should represent the whole model, with the measurement and structural ones. 

Reviewer 3 Report

Dear Authors,

Thank you for the opportunity to review the manuscript entitled ‘Pathways of neuroticism, social support, and sleep quality to antenatal depression during the third trimester of pregnancy’.

This manuscript reports findings from a cross-sectional survey with 773 women, examining direct and indirect associations between neuroticism, levels of social support, sleep quality and antenatal depression. Participants were recruited from one of three hospital sites in China. The findings provide further evidence for the role of dispositional traits (i.e., neuroticism) with antenatal depression, and for the role of perceived and objective social support and sleep in this context.

This manuscript is very well written, and was genuinely a pleasure to read. The findings presented are of value for considering methods to enhance social support during pregnancy, to mitigate the likelihood that antenatal depression occurs. My comments below are largely superficial suggestions just to enhance the presentation of this fantastic study.

Introduction

  • Page 1, lines 36 to page 2 line 45 includes a number of pivotal statements that require supporting with references
  • Page 2 line 48: ‘research’ not ‘researches’
  • Page 2 line 63: reference needed to support the notion that ‘in severe cases, even their social functions will also be affected’
  • Page 2 line 73: non-neuroticism is strange phraseology, may I suggest ‘lower levels of neuroticism’

General comments: given that antenatal depression and anxiety are often comorbid, could the researchers insert rationale for considering only depression in the present study? Similarly, could the rationale for considering the pathways between neuroticism, sleep, social support and antenatal depression in late pregnancy be enhanced? Arguably understanding these pathways earlier in pregnancy would afford greater opportunity for meaningful intervention.

Method

Setting and participants

  • Page 3 line 102: what was the process for random selection of the three sites? Were any parameters set to include sites that would support the aim of enhancing representativeness in the population recruited?
  • Page 3 line 106: ‘able to communicate normally’ could be re-worded to enhance clarity.
  • Was this an opportunity sample, or was recruitment consecutive?
  • Could the authors insert the final N as used for the analysis here too (i.e., 773/900 approached)

Instruments

  • The EPQ-N requires a reference (the original version). Has the EPQ-N previously been used with pregnant populations, or is there any indication of psychometric utility for the Chen’s Chinese version?
  • I think that clarity around the use of the SSRS could be enhanced. It is mentioned that the scale was developed with reference to existing tools and shaped for cultural utility. How many items, and what basis were items from existing tools selected? What was the process undertaken here?

Results

  • It is noted that the demographic characteristics of participants is presented in another paper, but there is no reference provided for readers to identify this information. I would suggest that including the demographic information in this manuscript too is essential to enable the reader to review participant characteristics in light of the findings
  • Page 5, subheading ‘The correlation between variables’ – this subsection could be streamlined to briefly summarise the bivariate associations as indicated in Table 2. A comment on the strength of these associations would also be welcome. I would also note again here that higher PSQI scores indicate worse sleep quality

Discussion

  • I would suggest beginning this section with an overview of the findings aligned with the aims/objectives of the paper, in addition to describing the participant characteristics in line with depression/ neuroticism.
  • Page 6 lines 250-253 (‘A study on…’) this repeats a sentence that was presented in the introduction. Use the discussion to interpret the findings in light of existing literature, moving on from information as presented earlier in the manuscript
  • It is correct that focusing on changeable factors relevant for antenatal depression is advisable, rather than targeting interventions at the level of disposition. However, perhaps the authors could expand upon the detail provided about how these findings can be used to inform/develop methods of improving social support. What are the key implications here?